# Improvement of the Approximation Accuracy of LED Radiation Patterns

**Ivan Rachev [1], Todor Djamiykov [2], Marin Marinov [2] and Nikolay Hinov [3],***

[1]  Department of Electronics, Faculty of Electronics and Automation, Technical University of Sofia,
    Plovdiv Branch, 25 Tsanko Diustabanov St., Plovdiv 4000, Bulgaria; ivr@tu-plovdiv.bg
[2]  Department of Electronics, Faculty of Electronic Engineering and Technologies, Technical University of
    Sofia 8, Kliment Ohridski Blvd., BG-1756 Sofia, Bulgaria; tsd@tu-sofia.bg (T.D.); mbm@tu-sofia.bg (M.M.)
[3]  Department of Power Electronics, Faculty of Electronic Engineering and Technologies, Technical University
    of Sofia 8, Kliment Ohridski Blvd., BG-1756 Sofia, Bulgaria
*   Correspondence: hinov@tu-sofia.bg; Tel.: +35929652569

**Abstract:** For the great variety of light-emitting diodes (LEDs), there exists a wide range of LED radiation patterns. An approach for constructing patterns of higher accuracy is here considered. The latter is required when the design of optoelectronic systems or their optimization is carried out analytically. A weighting function is introduced that allows increasing the gradient of the diagram of different widths. It has been selected through mathematical analysis of the emission diagrams of different LEDs used in optoelectronic systems. Based on the least squares method an algorithm is created, and programs are developed in MATLAB environment to estimate the parameters of the approximation function. Its accuracy is evaluated by comparison with the approximation with Lambert source of order n. The results show higher accuracy of the proposed approximation function compared to those obtained by conventional methods. Recommendations on the application of the proposed approach are given.

**Keywords:** approximation methods; LED lamps; mean square error methods; optical devices; optical design

## 1. Introduction

Nowadays, light-emitting diodes (LEDs) are everywhere, in different forms, and cover a wide range of applications from solid-state lighting to indicator lights. Because of their attractive characteristics, they are gradually taking over conventional light sources [1]. LEDs are more difficult to model than traditional sources. There exists a wide range of LED radiation patterns. A radiation pattern describes the relative light strength in any direction from the light source. Numerous approaches are used to model this in real light sources. The models currently employed can be classified as ray tracing or analytical approximations [2,3].

A ray-tracing model is very useful when analyzing and designing the package and the secondary optics for the source.

The analytic models are very useful when studying and optimizing the radiation transfer from the source to a target without intermediate complex optics. This is very common with LEDs because they have integrated optics. An analytic equation of the radiation pattern gives researchers and lighting designers more flexibility in analyzing the light in their applications [4–6].

One of the most important characteristics of light sources is the radiation pattern $P(\varphi, \vartheta)$ defined by the relationship:

$$P(\varphi, \vartheta) = \frac{I(\varphi, \vartheta)}{I_0} \leq 1 \tag{1}$$

where $I(\varphi, \vartheta)$ denotes the intensity of light emitted in a direction determined by the angles $\varphi$ and $\vartheta$ (in a spherical coordinate system), $I_0$ is the maximum intensity,

The intensity $I$ is here understood as the following ratio:

$$I = \frac{d\Phi}{d\Omega} \tag{2}$$

where $d\Phi$ is the elementary optical flux (optical power) in W and $d\Omega$ is the elementary spatial angle of the flux propagation in steradian.

The LED radiation and the associated radiation pattern are illustrated in Figure 1.

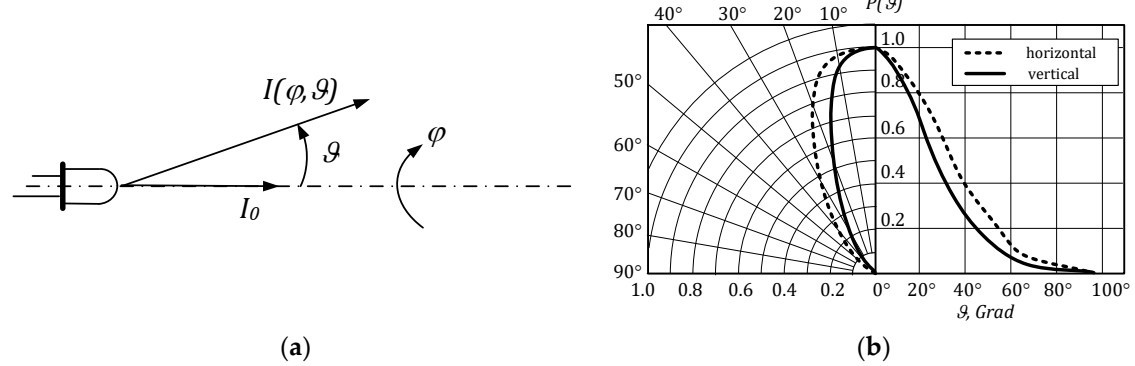

**Figure 1.** (**a**) Light-emitting diode (LED) radiation; (**b**) A typical LED radiation pattern.

In some commonly used light sources, such as LEDs, the radiation pattern is symmetric with respect to the optical axis. The maximum intensity in the diagram is then directed along with the optical axis, and the radiation pattern itself is a function only of the angle $\vartheta$, i.e.,

$$P(\varphi, \vartheta) = P(\vartheta), \tag{3}$$

This special case is here considered because it is of great importance for the practice.

The radiation pattern is given mostly in tabular or graphic form in catalogs. This is related to the measurement specificity. On the other hand, it is convenient to use an analytical approach in the design of optoelectronic systems and optimization of their parameters and structure. For this purpose, the radiation pattern is to be approximated. The most commonly used approximation function in a symmetric diagram is:

$$\hat{P}(\vartheta) = cos^{\hat{n}}(\vartheta) \tag{4}$$

That is, the light source is considered as a Lambert source of n-th order [7,8]; the exponent $\hat{n}$ is the estimation selected according to a given criterion.

A typical LED radiation pattern is shown in Figure 1b.

In some catalogs, the radiated flux and the radiation pattern are given in a standardized form. To determine the maximum intensity in this case, it is necessary to use Formulas (1)–(4). By means of mathematical transformations, we obtain:

$$\Phi = \int_0^{2\pi} d\varphi \int_0^{\frac{\pi}{2}} I_0 cos^n \vartheta sin\vartheta d\vartheta = 2\pi I_0 \frac{1}{n+1} \tag{5}$$

from which we get

$$I_0 = \frac{(n+1)\cdot\Phi}{2\pi} \leq 1 \tag{6}$$

The approximation of LED radiation patterns by (4) describes very well the physical processes and, respectively, radiation of flat surfaces. However, modern LEDs' housing is quite often transparent and performs a focusing function. This fact considerably changes the radiation pattern and is the reason other approximation functions are used as they can better describe the radiation pattern.

The present paper aims to study the accuracy of approximation of the LEDs radiation pattern. The task is to propose more accurate approximation functions when the source of light contains a focusing optical system. The aim is also to develop and study an algorithm for evaluation of the parameters of the approximation functions [9].

The precise determination of the radiation pattern will allow the correct specification of basic parameters (focal length, relative aperture, shadowing) of the optics which can be used to form the radiation from the semiconductor crystal. If the parameters of the additional optics used are known, the cumulative radiation pattern (diagram) will be more precisely determined, and the photometric calculations will be more accurate as well.

## 2. Analysis of Radiation Patterns

When the body of LEDs has focusing functions, the analysis of LEDs radiation patterns shows that to increase the accuracy of approximation the approximation Function (4) is to be corrected. It is appropriate to take it in the form of weighting function:

$$\hat{P}(\vartheta) = g(\vartheta, parameters) \cdot cos^{\hat{n}}(\vartheta) \tag{7}$$

The correction function $g(\vartheta, parameters)$ should meet the following requirements:

- It must be even since the pattern is symmetric to the optical axis;
- It must be normalized, i.e., $\lim_{\vartheta \to 0} g(\vartheta, parameters) = 1$ for any admissible value of the parameters (then the approximation Functions (4) and (7) become equal).

The function $\hat{g}(\vartheta, \hat{\beta}) = e^{-\frac{\vartheta^2}{\beta^2}}$ satisfies the above-listed requirements. When $\hat{\beta} \to \infty$ this function meets the second requirement and $\hat{P}(\vartheta) = e^{-\frac{\vartheta^2}{\beta^2}} \cdot cos^{\hat{n}} \vartheta$ becomes $\hat{P}(\vartheta) = cos^{\hat{n}} \vartheta$.

Hence, an appropriate LED radiation pattern (function) is:

$$\hat{P}(\vartheta) = e^{-\frac{\vartheta^2}{\beta^2}} \cdot cos^{\hat{n}} \vartheta \tag{8}$$

Using this function, it is possible, irrespective of the different nature of the approximated signals, to apply an approach similar to the one for determining the parameters [10]. To evaluate the optimal properties of the parameters, it is appropriate to use the criteria of minimal root mean square error (RMSE). It is assumed that the measured diagram $P$ and the approximating one $\hat{P}$ are discretized and presented as a set of the two samples $P(\vartheta_k)$ and $\hat{P}(\vartheta_k)$ at discrete angles $\vartheta_k = k \cdot \Delta\vartheta$, where $k = 0, 1, 2, \ldots, N$, and $\Delta\vartheta$ is the angle step of performed measurements of $P(\vartheta_k)$. To minimize RMSE, it is necessary:

$$E = \sum_{k=0}^{N} \left[ P(\vartheta_k) - \hat{P}(\vartheta_k) \right]^2 \to min,$$

That is, to find the minimum of

$$E = \sum_{k=0}^{N} \left[ P(\vartheta_k) - e^{-\left(\frac{\vartheta_k^2}{\beta^2}\right)} cos^{\hat{n}}(\vartheta_k) \right]^2. \tag{9}$$

The values of interest $\hat{\beta}$ and $\hat{n}$ are solutions of the following simultaneous equations:

$$\left|\begin{array}{l} D_{\hat{\beta}} = \frac{\partial E}{\partial \hat{\beta}} = 0 \\ D_{\hat{n}} = \frac{\partial E}{\partial \hat{n}} = 0 \end{array}\right. .$$

(10)

The parameter $\hat{\beta}$ can be determined from the first Equation (10):

$$D_{\hat{\beta}} = \sum_{k=0}^{N} 2 \left[ P(\vartheta_k) - e^{-\left(\frac{\vartheta_k^2}{\hat{\beta}^2}\right)} \cos^{\hat{n}}(\vartheta_k) \right] \left[ -\left(\frac{2\vartheta_k^2}{\hat{\beta}^3}\right) e^{-\left(\frac{\vartheta_k^2}{\hat{\beta}^2}\right)} \cos^{\hat{n}}(\vartheta_k) \right] = 0.$$

The expression within the second square brackets changes very slightly with the change of $\vartheta_k$ and, thus, it can be treated as a constant.

So, the final equation for $\hat{\beta}$ is:

$$D_{\hat{\beta}} = \sum_{k=0}^{N} \left[ P(\vartheta_k) - e^{-\left(\frac{\vartheta_k^2}{\hat{\beta}^2}\right)} \cos^{\hat{n}}(\vartheta_k) \right] = 0.$$

(11)

Equation (11) is non-linear with respect to $\hat{\beta}$, and its solution can be found using numerical methods. We use a graph-analytic approach. For that purpose, it is assumed that the true radiation pattern has the same form as (8), with true values of parameters $\beta = 20, deg$ and $n = 4$.

Figure 2 shows the graph of dependence $D_{\hat{\beta}}$ within the interval $\hat{\beta} = 0 - 100 deg$ and for $\hat{n} = 4$ and $\hat{n} = 8$. It can be seen that the function has no local extrema in this specific interval.

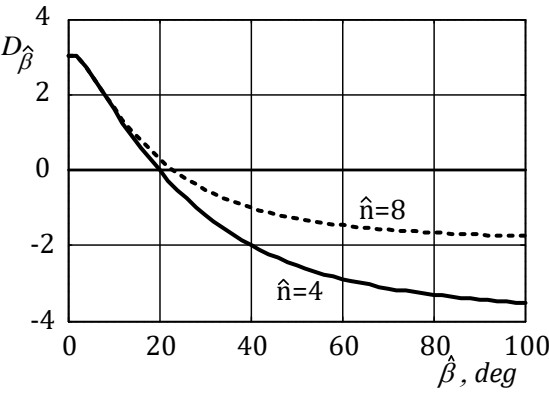

**Figure 2.** The dependence of $D_{\hat{\beta}}$ on $\hat{\beta}$ for $\hat{n} = 4$ and $\hat{n} = 8$.

When the rough estimate $\hat{n} = 8$ of the parameter $n$ is used, the solution of Equation (11) for the estimate $\hat{\beta}$ is not determined precisely and is quite different from the true value $\beta = 20$. The reason is that the algorithm "compensates" the wrong estimate $\hat{n} = 8$ with the parameter estimation $\hat{\beta}$ so that the root mean square error is minimal.

An analogous result can be obtained for the parameter $n$. After replacing (8) in (9) and then in the second Equation (10) it turns out that the estimate $\hat{n}$ is a solution to the equation:

$$D_{\hat{n}} = \sum_{k=0}^{N} \left[ P(\vartheta_k) - e^{-\left(\frac{\vartheta_k^2}{\hat{\beta}^2}\right)} . \cos^{\hat{n}}(\vartheta_k) \right] \left[ -e^{-\left(\frac{\vartheta_k^2}{\hat{\beta}^2}\right)} . \cos^{\hat{n}}(\vartheta_k) . \ln \cos(\vartheta_k) \right] = 0.$$

The expression within the second square brackets changes very slightly with the change of $\vartheta_k$ and, thus, it can be treated as a constant. So, the final equation for $\hat{n}$ is:

$$\sum_{k=0}^{N} \left[ P(\vartheta_k) - e^{-\left(\frac{\vartheta_k^2}{\hat{\beta}^2}\right)} . \cos^{\hat{n}}(\vartheta_k) \right] = 0.$$

(12)

In Figure 3 the graphs of $D_{\hat{n}}$ are presented when $\hat{n}$ changes in the range $\hat{n} = 1 - 10$. The true value of $n$ is $n = 4$. The broken line graph corresponds to $\hat{\beta} = 50$, and the continuous line—to $\hat{\beta} = \beta = 30$. It can be seen that if $\hat{\beta}$ is accurately evaluated, the solution of (12) matches the true value of the parameter: $\hat{n} = n$. In the case of a rough estimate $\hat{\beta}$, for instance $\hat{\beta} = 50$, the estimation $\hat{n}$ differs substantially from the true value (obtained $\hat{n} \approx 8,5$ rather than 4), whereby the algorithm minimizes the error (9).

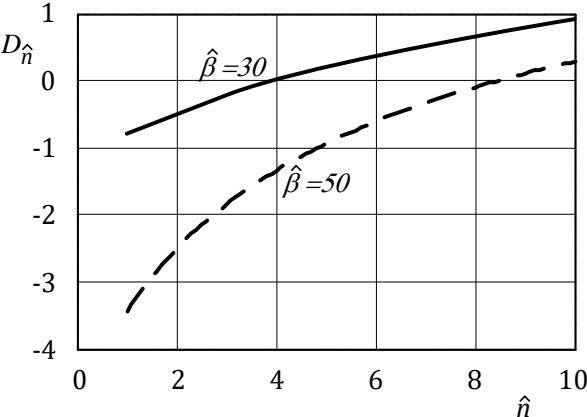

**Figure 3.** Graphs of $D_{\hat{n}}$ depending on $\hat{n}$ for $\hat{\beta} = 30$ and $\hat{\beta} = 50$.

From (11) and (12) it follows that $D_{\hat{n}} = D_{\hat{\beta}}$. Therefore, the Newton-Raphson method for numerical solution of systems of nonlinear equations cannot be applied [11,12]. The estimated parameters $\hat{\beta}$, $\hat{n}$ of the approximate diagram $\hat{P}(\vartheta, \hat{\beta}, \hat{n})$ are iteratively determined: In calculation of each of them, the other one is considered constant. The calculations are terminated when the minimum error (9) is achieved with a given accuracy.

Since the parameter $\beta$ is less dependent on the parameter $n$, the latter is calculated by the nodal approximation method. For the calculation of the estimated value $\hat{\beta}$, the following iterative formula can be used:

$$\hat{\beta}_{k+1} = \hat{\beta}_k - \frac{D_{\hat{\beta}}}{G_{\hat{\beta}}}, \tag{13}$$

where $G_{\hat{\beta}}$ is the partial derivative of $D_{\hat{\beta}}$ with respect to $\hat{\beta}$:

$$G_{\hat{\beta}} = \frac{\partial D_{\hat{\beta}}}{\partial \hat{\beta}} = -\frac{2}{\hat{\beta}} \cdot \sum_{k=0}^{N} \left[ e^{-\left(\frac{\vartheta_k^2}{\hat{\beta}^2}\right)} \cdot \cos^{\hat{n}}(\vartheta_k) \right] = 0. \tag{14}$$

The iterative process is terminated when the inequality $\left| \hat{\beta}_{j+1} - \hat{\beta}_j \right| \leq \varepsilon_\beta$ is satisfied; $\varepsilon_\beta$ is the admissible error.

## 3. About the Algorithm for Calculation of the Parameters

The algorithm used to determine the parameters of the approximation radiation pattern is shown in Figure 4.

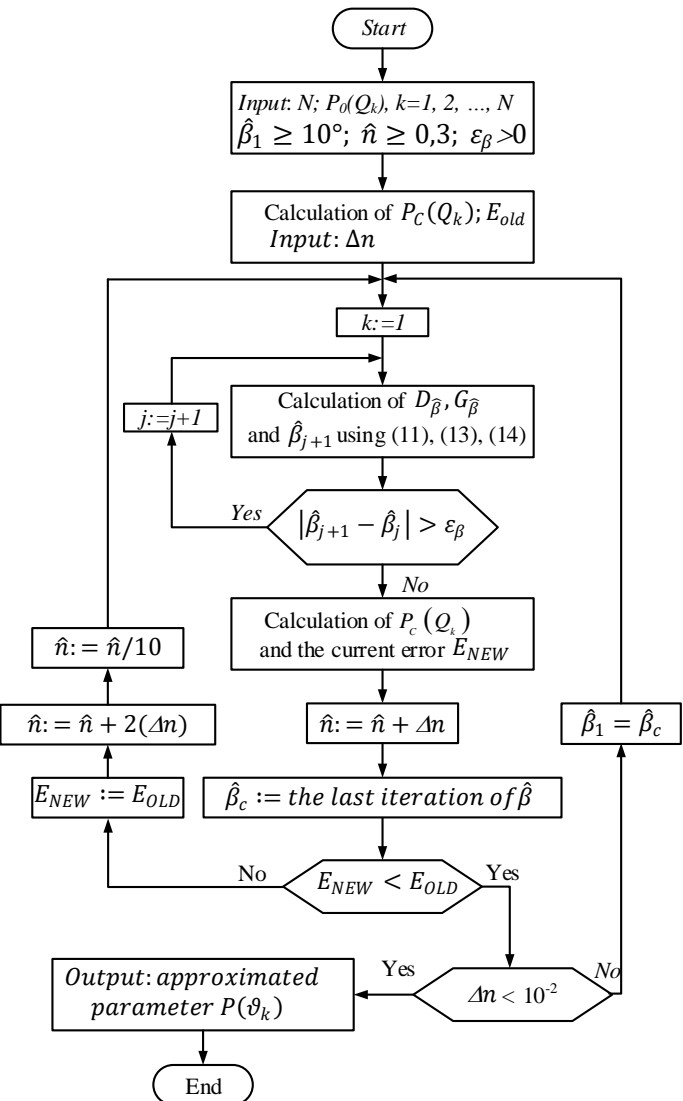

**Figure 4.** Algorithm for parameters determination of the approximation radiation pattern.

## 4. Validation of the Proposed Algorithm

### 4.1. Parameter Evaluation

For testing the developed algorithm for the iterative method, a software program was written in the MATLAB environment. The test was performed using the following input data:

(1) The radiation pattern was of the type:

$$P(\vartheta) = e^{-\left(\frac{\vartheta}{\beta}\right)} \cdot cos^n(\vartheta). \tag{15}$$

(2) The true values of the parameter where: $\beta = 60.0$ and $n = 3.10$.

(3) The angle $\vartheta$ changes from $0°$ to $90°$, the step $\Delta\vartheta = 4.5°$ and $N = 21$.

Since the parameter $\beta$ changes in a larger interval than the other parameter $n$, the value of $\hat{\beta}$ is calculated to the first decimal place, and the estimate $\hat{n}$ is calculated to the second decimal place.

The estimated parameters are in a good coincidence with the true ones, i.e., they satisfy the required accuracy. Figure 5a shows a good level of matching of the true and approximated radiation pattern as well.

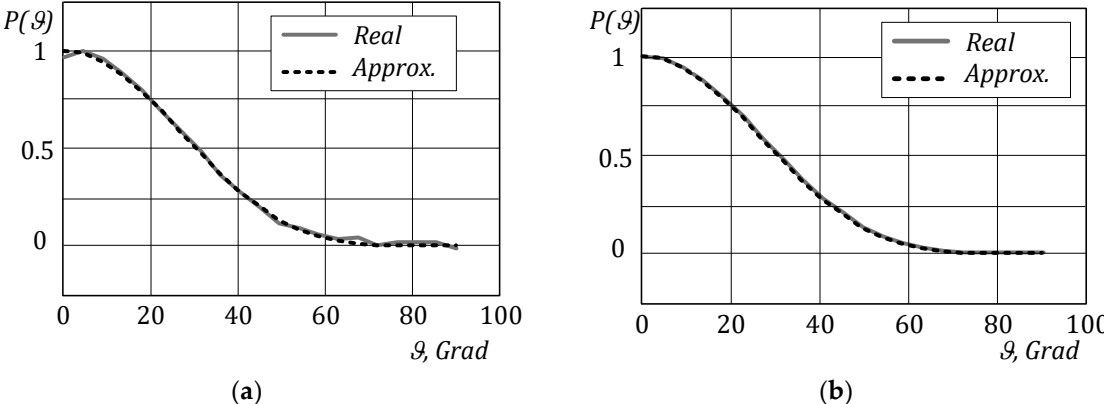

**Figure 5.** Approximation: (**a**) in case of no random measurement errors; (**b**) in case of random measurement errors.

### 4.2. Resistance of the Algorithm

The resistance of the algorithm to wrong values of $P(\vartheta_k)$ obtained by measuring was also tested. The errors in these values are simulated by taking into account (adding) the Gauss's noise into the radiation pattern (15). These errors in these values are simulated by an additive Gaussian noise with levels up to 1% of the measured values (SNR = 40 dB).

Figure 5b illustrates a good coincidence of the graphs of the real and approximated radiation pattern: It is a result of the "smoothing" effect of the approximation.

This test has proved the resistance of the proposed algorithm to deviations in case of the wrong measurement of the diagram. It should be noted that the big differences between the measured and the approximation values at the points of measurement of the radiation pattern are the reason for the error between the values of the parameters and their estimates increase according to (see below in Section 4.3) formula (16).

### 4.3. Root Mean Square Error

The third test aimed at the approximation of real LEDs radiation patterns and comparison of the root mean square error (RMSE) of the approximating Functions (4) and (8).

$$RMSE = \sqrt{\frac{\sum_{k=0}^{N}\left[P(\vartheta_k) - \hat{P}(\vartheta_k)\right]^2}{N}}. \tag{16}$$

We have chosen infra-red LED C3535SIRC-2B and L-53SF7C because they have wide and narrow radiation pattern, respectively. The calculations were carried out with both approximation Functions (4) and (8) and $\Delta\vartheta = 5°$. The RMSE of the two approximations is presented in Table 1.

**Table 1.** RMSE of the two approximations

| LEDs | RMSE | |
| --- | --- | --- |
| | $\hat{P}(\vartheta)=cos^{\hat{n}}(\vartheta)$ | $\hat{P}(\vartheta)=e^{-\left(\frac{\vartheta^2}{\beta^2}\right)}\cdot cos^{\hat{n}}(\vartheta)$ |
| LED C3535SIRC-2B | 0.0329 | 0.0118 |
| L-53SF7C | 0.0354 | 0.0332 |

Figure 6 shows the measured and the approximated radiation patterns of the two LEDs. The diagrams of C3535SIRC-2B with approximation Functions (4) and (8) are presented in Figure 6a,b, respectively, and of IR LED L-53SF7C—in Figure 6c,d respectively.

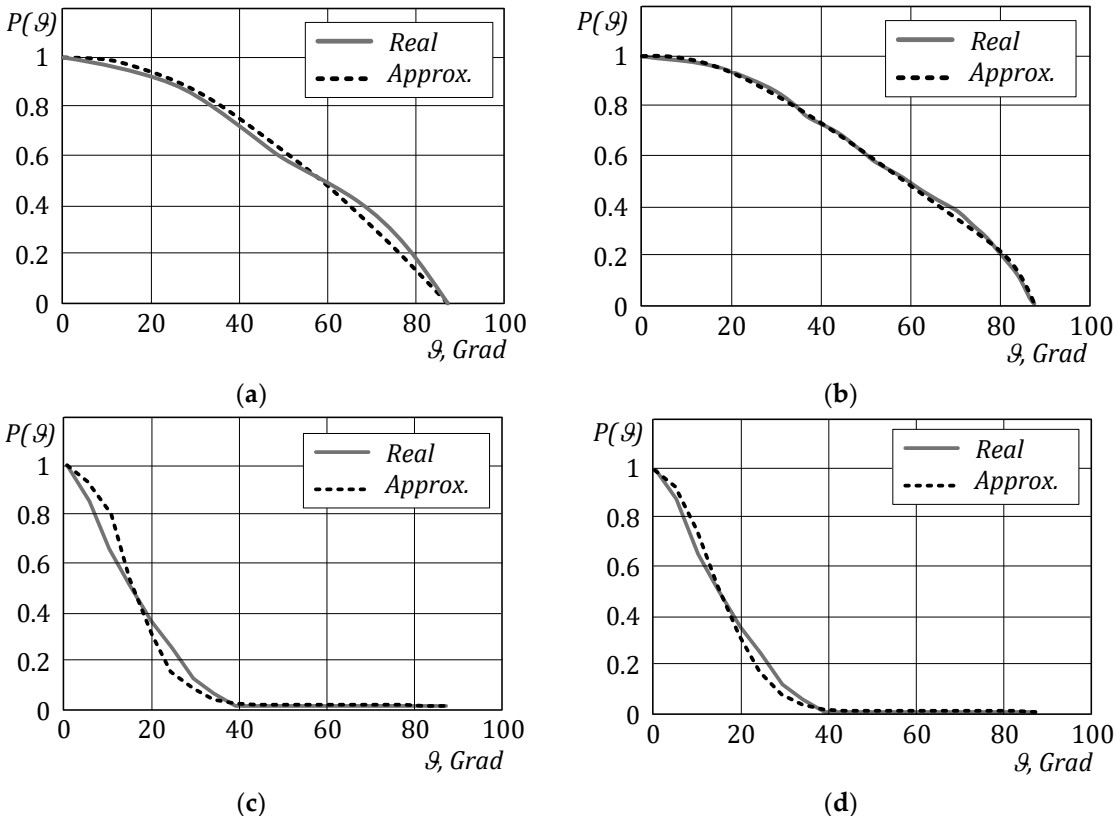

**Figure 6.** Approximation of the radiation diagrams: (**a,b**) IR LED C3535SIRC; (**c,d**) IR LED 53F7C, respectively with Functions (4) and (8).

## 5. Conclusions

The proposed approximation function provides an opportunity for reduction of the approximation error. This has been confirmed by the results in Table 1 as well as by experiments conducted with other LEDs. The achieved accuracy is no less than the corresponding one obtained with the approximation Function (4).

- The decrease of the error depends on the measured LED radiation pattern. The error decreases less when the true pattern is well described by the Function (4). Then the algorithm calculates so large $\hat{\beta}$ that the Gaussian function is practically irrelevant. In addition, for narrower patterns, the increase in accuracy is less, as the overall error, in this case, is determined by the error in the measurement of the diagram (the weight of each point is large);
- The proposed algorithm works correctly and allows the determination of the estimated parameters $\hat{\beta}$ and $\hat{n}$ effectively.

As recommendations regarding the application of the proposed approximation function, we wish to point out that in the design of the energy balance of an optoelectronic system working at long distances and in the open air, the error in estimating the intensity can be bigger; then we can use Function (4) to reduce the complexity of the engineering calculations.

For the optoelectronic system, it is important to know the emitted intensity at every angle (this is the case of multi-beam interferometers), the proposed approach can significantly improve the accuracy, and the proposed algorithm for estimation of the parameters can be used [13–15].

**Author Contributions:** I.R., T.D., M.M. and N.H. were involved in the full process of producing this paper including modelling, data processing, and preparing the manuscript.

**Funding:** This work was supported by the Bulgarian Ministry of Education and Science under the National Research Programme "Low carbon energy for transport and live", approved by DCM # 577/17.08.2018.

**Conflicts of Interest:** The authors declare no conflict of interest.

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
