# Peer review of "Improvement of the Approximation Accuracy of LED Radiation Patterns"

_electronics, doi:10.3390/electronics8030337_

Round 1
Reviewer 1 Report
In this manuscript, the authors developed an algorithm to calculate the LED radiation pattern and demonstrated higher accuracy of the proposed approximation function compared to conventional methods. This paper is well presented and demonstrated interesting result, I think it is suitable for publication in Electronics.
Author Response
First of all we would like to thank you for the thorough review of our paper (electronics-459863) and the useful remarks to improve it.
Reviewer 1
Comments to the Authors
In this manuscript, the authors developed an algorithm to calculate the LED radiation pattern and demonstrated higher accuracy of the proposed approximation function compared to conventional methods.
This paper is well presented and demonstrated interesting result, I think it is suitable for publication in Electronics.
To Reviewer 1:
Thank you for your review and the expressed opinion, that the paper is suitable for publication in Electronics.
Reviewer 2 Report
The author proposes a approximation methods of LED radiation pattern with improved accuracy. (1) For a the practical application of LED, optics are usually used for either focusing or diffusing. Could the author comments the effects of optics and its influence on the current proposed model?
(2) There are lots of typos in the manuscript. For example, line 89, 120, 144, 147, 169... Please double check and correct them.
Author Response
Reviewer 2
Answers to reviewers
First of all we would like to thank you for the thorough review of our paper (electronics-459863) and the useful remarks to improve it.
Comments to the Authors
The author proposes an approximation method of LED radiation pattern with improved accuracy.
(1) For the practical application of LED, optics are usually used for either focusing or diffusing. Could the author comments the effects of optics and its influence on the current proposed model?
(2) There are lots of typos in the manuscript. For example, line 89, 120, 144, 147, 169...Please double check and correct them.
To Reviewer 2:
Thank you for your review of our paper (electronics-459863) and the valuable recommendations.
About
(1) …
The following comment concerning optics has been inserted below line 73 in the Introduction:
The precise determination of the radiation pattern will allow the correct specification of basic parameters (focal length, relative aperture, shadowing) of the optics which can be used to form the radiation from the semiconductor crystal. If the parameters of the additional optics used are known, the cumulative radiation pattern (diagram) will be more precisely determined and the photometric calculations will be more accurate as well.
About
(2) …
We did our best to correct the typos using track changes. The revised version of the paper is uploaded to the editorial system.
Reviewer 3 Report
This paper is about achieving high accuracy of LED radiation patterns by a specific algorithm, and further evaluated by comparison with the approximation with Lambert source of order n to prove that a higher accuracy is achieved. It is novel and the logic is reasonable. So I recommend it to be accepted.
Author Response
Answers to reviewers
First of all we would like to thank you for the thorough review of our paper (electronics-459863) and the useful remarks to improve it.
Reviewer 3
Comments to the Authors
This paper is about achieving high accuracy of LED radiation patterns by a specific algorithm, and further evaluated by comparison with the approximation with Lambert source of order n to prove that a higher accuracy is achieved. It is novel and the logic is reasonable.
So I recommend it to be accepted.
To Reviewer 3:
Thank you for your review and recommendation the paper (electronics-459863) to be accepted for publication in Electronics.
Round 2
Reviewer 2 Report
The authors have addressed all my concerns. I recommend acceptance of the manuscript.